# A novel selective PPARα modulator, pemafibrate promotes ischemia-induced revascularization through the eNOS-dependent mechanisms

Hiroshi Kawanishi[1], Koji Ohashi [2]*, Hayato Ogawa[1], Naoya Otaka[1], Tomonobu Takikawa[1], Lixin Fang[1], Yuta Ozaki[1], Mikito Takefuji[1], Toyoaki Murohara[1], Noriyuki Ouchi[2]*

1 Department of Cardiology, Nagoya University Graduate School of Medicine, Nagoya, Japan, 2 Department of Molecular Medicine and Cardiology, Nagoya University Graduate School of Medicine, Nagoya, Japan

* ohashik@med.nagoya-u.ac.jp (KO); nouchi@med.nagoya-u.ac.jp (NO)

**Data Availability Statement:** All relevant data are within the paper and its Supporting Information files.

## Abstract

### Objective

Cardiovascular disease is a leading cause of death worldwide. Obesity-related metabolic disorders including dyslipidemia cause impaired collateralization under ischemic conditions, thereby resulting in exacerbated cardiovascular dysfunction. Pemafibrate is a novel selective PPARα modulator, which has been reported to improve atherogenic dyslipidemia, in particular, hypertriglyceridemia and low HDL-cholesterol. Here, we investigated whether pemafibrate modulates the revascularization process in a mouse model of hindlimb ischemia.

### Methods and results

Male wild-type (WT) mice were randomly assigned to two groups, normal diet or pemafibrate admixture diet from the ages of 6 weeks. After 4 weeks, mice were subjected to unilateral hindlimb surgery to remove the left femoral artery and vein. Pemafibrate treatment enhanced blood flow recovery and capillary formation in ischemic limbs of mice, which was accompanied by enhanced phosphorylation of endothelial nitric oxide synthase (eNOS). Treatment of cultured endothelial cells with pemafibrate resulted in increased network formation and migratory activity, which were blocked by pretreatment with the NOS inhibitor $N^G$-nitro-L-arginine methyl ester (L-NAME). Pemafibrate treatment also increased plasma levels of the PPARα-regulated gene, fibroblast growth factor (FGF) 21 in WT mice. Systemic administration of adenoviral vectors expressing FGF21 (Ad-FGF21) to WT mice enhanced blood flow recovery, capillary density and eNOS phosphorylation in ischemic limbs. Treatment of cultured endothelial cells with FGF21 protein led to increases in endothelial cell network formation and migration, which were canceled by pretreatment with L-NAME. Furthermore, administration of pemafibrate or Ad-FGF21 had no effects on blood flow in ischemic limbs in eNOS-deficient mice.

**Funding:** This work was supported by Grant-in-Aid for Scientific Research (2620H00571 to N.O., 2617H04175 to N.O., 2616K09512 to K.O.); grants from Takeda Science Foundation (2600007594 and 2600007051 to N.O); and a grant from the Suzuken Memorial Foundation (2600007887 to N.O.).

**Competing interests:** The authors have declared that no competing interests exist.

## Conclusion

These data suggest that pemafibrate can promote revascularization in response to ischemia, at least in part, through direct and FGF21-mediated modulation of endothelial cell function. Thus, pemafibrate could be a potentially beneficial drug for ischemic vascular disease.

## Introduction

Cardiovascular disease is a major cause of death worldwide [1]. Obesity-related metabolic disorders including type 2 diabetes and dyslipidemia contribute to impaired collateralization and vascular insufficiency under ischemic conditions, thereby leading to exacerbation of cardiac dysfunction and tissue injury [2–4]. Thus, the enhancement of collateral vessel development can be a promising therapeutic target of cardiovascular diseases.

Peroxisome proliferator-activated receptor (PPAR) α is a member of the nuclear hormone receptor superfamily of ligand-activated transcription factors, and has an important effect on lipid and lipoprotein metabolism [5]. PPARα agonists decrease plasma triglyceride levels and increase plasma high density lipoprotein (HDL)-cholesterol levels. In addition, PPARα agonists have various roles in regulation of cardiovascular homeostasis such as angiogenesis [6–8]. On the other hand, existing PPARα agonists, such as fenofibrate and bezafibrate, sometimes cause adverse effects, especially for use in patients with renal dysfunction or for concomitant use of statin [9–11].

Pemafibrate is a novel selective PPARα modulator (SPPARMα), which has a higher PPARα agonistic activity and selectivity than existing PPARα agonists. Recent evidence indicates that pemafibrate has strong effects on lowering triglycerides and improving atherogenic dyslipidemia without a significant increase in adverse events even in the patients receiving statins [12, 13]. Thus, it is conceivable that pemafibrate can be a useful drug to reduce cardiovascular risk. In this regard, it has been reported that pemafibrate administration improves dyslipidemia and reduces atherosclerotic lesion formation in a mouse model of atherosclerosis [14]. However, little is known about the effect of pemafibrate on the development of ischemic vascular diseases. In the present study, we investigated whether pemafibrate modulates revascularization process in a mouse model of hindlimb ischemia.

## Materials and methods

### Ethics statement

All animal study protocols were approved by the Institutional Animal Care and Use Committee in Nagoya University.

### Materials

Pemafibrate was kindly provided by Kowa Co. Ltd (Nagoya, Japan). Mouse CD31 antibody was purchased from BD Pharmingen (San Jose, CA)(550274). Antibodies of phosphorylated eNOS (Ser-1177)(9571), eNOS (32027) and Tubulin (2144) were purchased from Cell Signaling Technology (Beverly, MA). *N*G-nitro-l-arginine methyl ester (L-NAME) was purchased from Sigma (St. Louis, MO) (N5751). GW6471 was purchased from Cayman Chemical (11697). Recombinant human FGF21 protein was purchased from R&D system (2539-FG-025). Plasma FGF21 levels were measured by ELISA kit (R & D system)(MF2100) [15]. Plasma adiponectin levels were determined by ELISA kit (Otsuka Pharmaceutical Co. Ltd.)(410713).

Adenoviral vectors expressing mouse full-length FGF21 (Ad-FGF21) were constructed under the control of the CMV promoter [16, 17]. Adenoviral vectors expressing β-galactosidase (Ad-βgal) were used as controls [18]. Lipid profiles and plasma glucose were analyzed by enzymatic kits (Wako Pure Chemical Industries, Ltd) (total cholesterol, 439–17501) (triglyceride, 290–63701) (glucose, 439–90901).

## Mouse model of hindlimb ischemia

Male wild-type (WT) or eNOS-knockout (eNOS-KO) (Jackson Laboratory) mice at the ages of 6 weeks were fed normal diets containing pemafibrate (0.12 mg/kg/day) or vehicle (Control) for 8 weeks. At the age of 10 weeks, WT or eNOS-KO mice were subjected to unilateral hind limb surgery to remove the left femoral artery and vein under anesthesia [19–22]. In some experiments, Ad-βgal at $1×10^9$ plaque-forming units (pfu) or Ad-FGF21 at $1×10^9$ pfu was intravenously injected into right jugular vein 3 days prior to the surgery as previously described [19, 23]. Hindlimb blood flow was measured by a laser Doppler blood flow analyzer (Moor LDI, Moor Instruments) immediately before surgery and on postoperative days 3, 7, 14 and 28. To avoid data variations caused by ambient light and temperature, hind limb blood flow was expressed as the ratio of left (ischemic) to right (non-ischemic) LDBF. Capillary density within thigh adductor muscle was analyzed by immunohistochemistry [19, 23]. Muscle samples were embedded in OCT compound (Miles, Elkhart, IN) and snap-frozen in liquid nitrogen. Tissue slices (5 μm in thickness) were stained with anti-CD31 antibodies (BD Pharmingen). Fifteen randomly chosen microscopic fields from three different sections in each tissue block were examined for the presence of CD31-positive capillary endothelial cells. Capillary density was expressed as the number of CD31-positive cells per muscle fiber.

## Quantification of mRNA levels

Gene expression levels were quantified by real-time PCR method. Total RNA was extracted from skeletal muscle tissues, liver and HUVECs using RNeasy Mini Kit (Qiagen). RNA which had an OD260/280 ratio of 1.8 or greater was used for reverse transcription reaction. cDNA was produced from 0.5 μg total RNA using a Revatra Ace (Toyobo) [24]. PCR was performed with a Bio-Rad real-time PCR detection system using THUNDERBIRD SYBR qPCR Mix as a double-standard DNA-specific dye. Primers were 5'-GCTCCAAGCAGATGCAGCA-3' and 5'-CCGGATGTGAGGCAGCAG-3' for mouse 36B4, 5'-GCTGCTGGAGGACGGTTACA-3' and 5'-CACAGGTCCCCAGGATGTTG-3' for mouse FGF21, 5'-GCCCAGCAACATTATCCAGT-3' and 5'-GGTCAGACTTCCTGCTACGC-3' for mouse LPL, 5'-CGGAGTCCGGGCAGGT-3' and 5'-GCTGGGTAGAGAATGGATGAACA-3' for mouse TNF-α, 5'-GCTACCAAACTGGATATAATCAGGA-3' and 5'-CCAGGTAGCTATGGTACTCCAGAA-3' for mouse IL6, 5'-GCCTGTGTTTTCCTCCTTGC-3' and 5'-CTGCCTAATGTCCCCTTGA-3' for mouse IL1β, 5'-CCACTCACCTGCTGCTACTCAT-3' and 5'-TGGTGATCCTCTTGTAGCTCTCC-3' for mouse MCP1 and 5'-AGGTTGGATGGCAGGC-3' for mouse adiponectin. All results were normalized to 36B4.

## Cell culture

Human umbilical endothelial cells (HUVECs) were cultured in endothelial cell growth medium 2 (Lonza)(EBM-2 (CC-3156), EGM-2 (CC-4176)). HUVECs were cultured in the presence or absence of pemafibrate (10 nM) or recombinant FGF21 protein (10 nM) for the indicated lengths of time.

## Assessment of endothelial cell function

The formation of vascular-like structures by HUVECs on growth factor-reduced Matrigel (BD Biosciences) was performed as previously described (20,27). Differentiation was quantified by measuring the area of the "tube-like" networks that form in three randomly chosen fields from each well. Each experiment was repeated three times. Chemotaxis of HUVECs was assessed by transwell assay with polycarbonate membranes coated with fibronectin (Corning)(3415) [25]. HUVECs were added to the upper chamber, and serum-deprived media supplemented with pemafibrate, FGF21 or vehicle was added to the lower chamber. Cells were allowed to migrate through the pores of the membrane for 10 hours. Cell proliferation was assessed by MTS-based assay (Promega)(G3580) [26]. HUVECs were stimulated with pemafibrate, FGF21 or vehicle for the indicated lengths of time under normoxic or hypoxic condition. Hypoxic conditions were generated using an AnaeroPack (5% $O_2$, 5% $CO_2$, Mitsubishi GAS Chemical) (3276LJ).

## Western blot analysis

Tissue samples were homogenized in lysis buffer containing 1 mM PMSF (Cell Signaling Technology). Immunoblot analysis was performed with antibodies at a 1:1000 dilution, followed by incubation with a secondary antibody conjugated with horseradish peroxidase at a 1:5000 dilution. An ECL Prime Western blotting detection kit (GE healthcare) was used.

## Statistical analysis

Data are presented as mean ± S.E. The differences between two groups for variables with normal distributions were evaluated by unpaired Student's t-test. Differences between three or more groups were evaluated using one-way analysis of variance, with a post-hoc Tukey's test. A $P$ value < 0.05 denoted the presence of a statistically significant difference. All statistical analyses were performed using SPSS version 18.

# Results

## Pemafibrate enhances ischemia-induced revascularization in vivo

To examine the effects of pemafibrate on revascularization in response to ischemia, WT mice were fed normal chow diets containing pemafibrate or vehicle (control) followed by subjection to hindlimb ischemia surgery. Fig 1A shows representative laser Doppler blood flow (LDBF) images of hindlimb blood flow before surgery, after surgery, at day 14 and at day 28 after surgery. Pemafibrate administration significantly increased blood flow recovery in ischemic limbs at day 3, 7, 14, 21 or 28 after operation compared with control (Fig 1B).

To assess the extent of revascularization at a microcirculatory level, capillary density in non-ischemic or ischemic adductor muscles was assessed by staining with anti-CD31 antibody. Administration of pemafibrate significantly increased the number of CD31-positive cells in ischemic limbs of WT mice (Fig 1C). In contrast, pemafibrate did not affect the number of CD31-positive cells in non-ischemic limbs of WT mice. These findings indicate that pemafibrate promotes ischemia-induced revascularization in vivo. In addition, treatment with pemafibrate significantly reduced plasma triglyceride concentration compared with control, whereas no differences were observed in total cholesterol and glucose levels between two groups (Fig 1D). Treatment with pemafibrate also increased mRNA expression of lipoprotein lipase (LPL), which is a downstream molecule of PPARα, in the liver (S1 Fig).

To test whether pemafibrate directly affects endothelial cell function, HUVECs were placed on a Matrigel matrix and treated with pemafibrate or vehicle. Treatment of HUVECs with

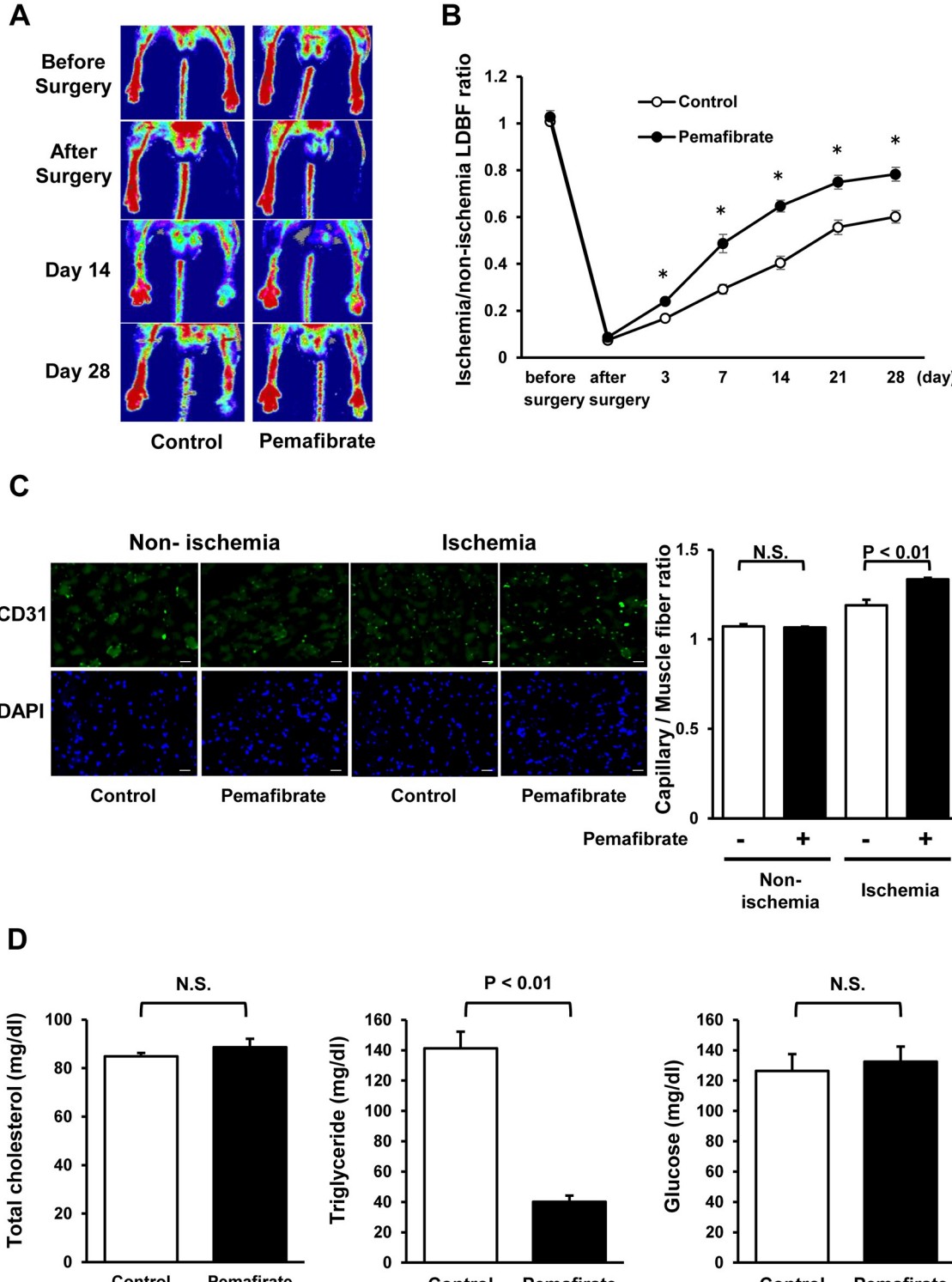

**Fig 1. Pemafibrate promotes blood flow recovery and capillary density in ischemic limbs of WT mice. A and B**. Effect of pemafibrate administration on blood flow recovery in ischemic limbs of WT mice. Representative Laser Doppler Blood Flow (LDBF) images of limb blood flow in WT mice fed diets containing pemafibrate or vehicle (control) are shown in Fig 1A. Quantitative analysis of the ischemic/non-ischemic LDBF ratio of WT mice receiving control or pemafibrate diet is shown in Fig 1B. *P<0.05. N = 8 in each group. **C**. Effect of pemafibrate administration on capillary density in ischemic limbs of WT mice. Representative immunostaining of ischemic muscle tissues with anti-CD31 antibody (green) and DAPI (blue) on postoperative day 28. Right panel

shows quantitative analyses of capillary density in ischemic muscles of control or pemafibrate-treated WT mice on postoperative day 28. N = 6 in each group. Scale bars show 50 μm. **D**. Plasma concentration of total cholesterol, triglyceride and glucose of control or pemafibrate-treated WT mice on postoperative day 28. N = 5 in each group (total cholesterol and triglyceride). N = 13 in each group (glucose).

pemafibrate increased network areas compared with vehicle (Fig 2A). Treatment with pemafibrate also enhanced migration and proliferation of HUVECs (Fig 2B and 2C). The stimulatory effects of pemafibrate on network formation and migration of HUVECs were canceled by GW6471, which is a specific inhibitor of PPARα (Fig 3B and 3C), indicating that pemafibrate can directly modulate endothelial behavior in a PPARα dependent manner. Treatment of HUVECs with pemafibrate also increased network areas, migration and proliferation compared with vehicle under hypoxic condition (S2 Fig).

Because eNOS is a key regulator of endothelial cell function [27], we evaluated whether pemafibrate regulates eNOS phosphorylation in HUVECs. Treatment of HUVECs with pemafibrate significantly increased phosphorylation levels of eNOS compared with vehicle culture (Fig 3A). To examine whether pemafibrate enhances endothelial cell function through the eNOS signaling pathway, HUVECs were pretreated with the NOS inhibitor, L-NAME followed by stimulation with pemafibrate or vehicle. Pretreatment with L-NAME abolished pemafibrate-induced enhancement of network formation and migration of HUVECs (Fig 3B and 3C). Consistently, pemafibrate treatment significantly increased phosphorylation levels of eNOS in ischemic adductor muscle, but not in non-ischemic muscle in WT mice compared with control at day 7 and day 28 after surgery (S3 Fig, Fig 3D). Furthermore, pemafibrate treatment had no significant effects on blood flow recovery in eNOS-KO mice throughout the experimental period (Fig 3E). These data suggest that pemafibrate promotes angiogenic responses in vitro and in vivo through the eNOS-dependent mechanism.

Because FGF21 acts as a target gene of PPARα with vasculo-protective effects [28], plasma concentration of FGF21 was measured in control and pemafibrate-treated WT mice. Treatment with pemafibrate robustly increased circulating levels of FGF21 compared with control at day 7 and day 28 after surgery (S4 Fig, Fig 4A). Concomitantly, hepatic expression of FGF21 was significantly higher in pemafibrate-treated WT mice than in control WT mice, whereas no significant difference in FGF21 mRNA levels in ischemic and non-ischemic skeletal muscle was observed between control and pemafibrate-treated mice at day 7 and day 28 after surgery (Fig 4B, S5A and S5B Fig). Pemafibrate treatment did not affect the expression of FGF21 in HUVECs (S5C Fig). In contrast, pemafibrate did not affect circulating levels of the vasculo-protective adipokine adiponectin (S6A Fig). Similarly, pemafibrate had no effects on mRNA expression of adiponectin in epidydimal fat tissue (S6B Fig).

To evaluate whether FGF21 modulates angiogenic response in vivo, adenoviral vectors expressing FGF21 (Ad-FGF21) or control vector (Ad-βgal) were intravenously injected into WT mice 3 days prior to surgery. Systemic administration of Ad-FGF21 significantly increased plasma FGF21 concentration in WT mice compared with Ad-βgal treatment at day 3 and day 17 after adenoviral vector administration (Fig 4C, S7A Fig). Ad-FGF21 administration also increased the expression of FGF21 in ischemic skeletal muscle of WT mice compared with Ad-βgal treatment at day 3 after adenoviral vector injection (S7B Fig).

Ad-FGF21 administration significantly enhanced blood flow recovery in ischemic limbs of WT mice compared with Ad-βgal treatment (Fig 4D). The number of CD31-positive cells was significantly higher in Ad-FGF21-treated mice compared with Ad-βgal-treated mice (Fig 4E). Furthermore, treatment with FGF21 protein promoted network formation, migration and proliferation of HUVECs compared with vehicle (Fig 5A, 5B and 5C). These results indicate

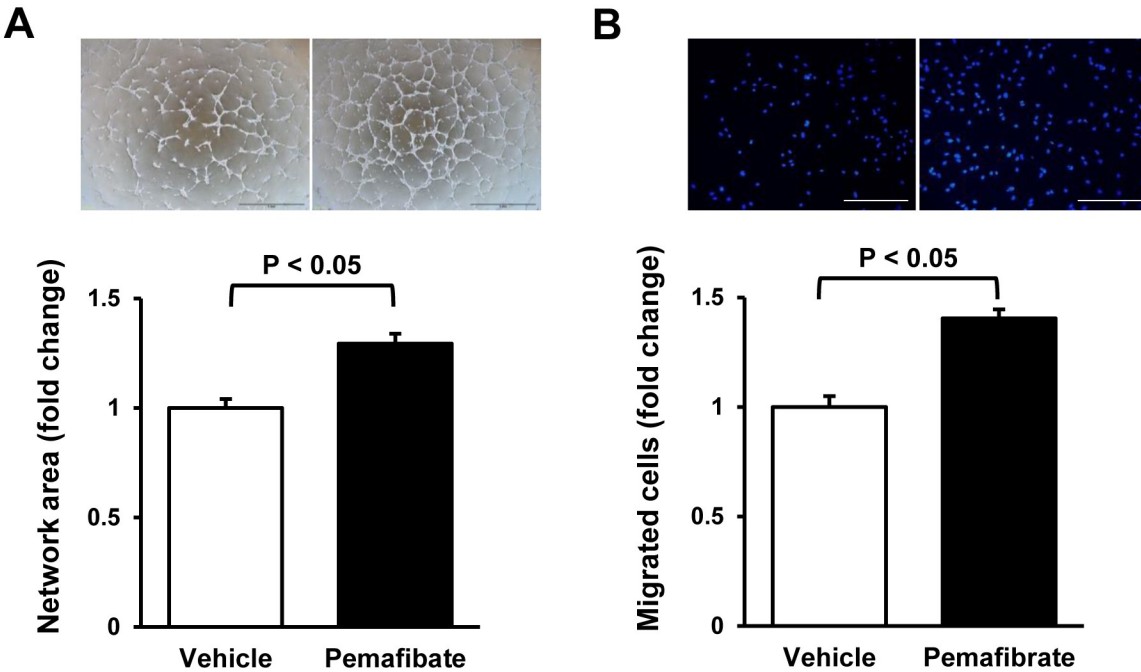

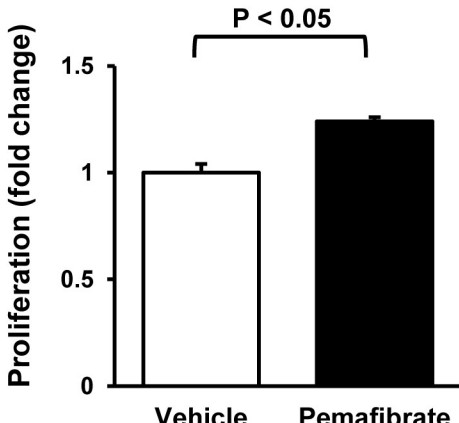

**Fig 2. Pemafibrate promotes endothelial cell function in vitro. A**. Endothelial cell network formation after treatment with pemafibrate. Upper panels show the representative photos of network formation of HUVECs at 16 h after treatment with pemafibate (10 nM) or vehicle. Lower panel shows the quantitative analysis of network area. N = 4 in each group. Scale bars show 1 mm. **B**. The number of migrated HUVECs at 8 h after treatment with pemafibrate (10 nM) or vehicle. Upper panels show the representative photos of DAPI staining of migrated HUVECs. N = 6 in each group. Scale bars show 200 μm. **C**. Proliferative activity of HUVECs at 24 h after treatment with pemafibrate (10 nM) or vehicle. N = 8 in each group.

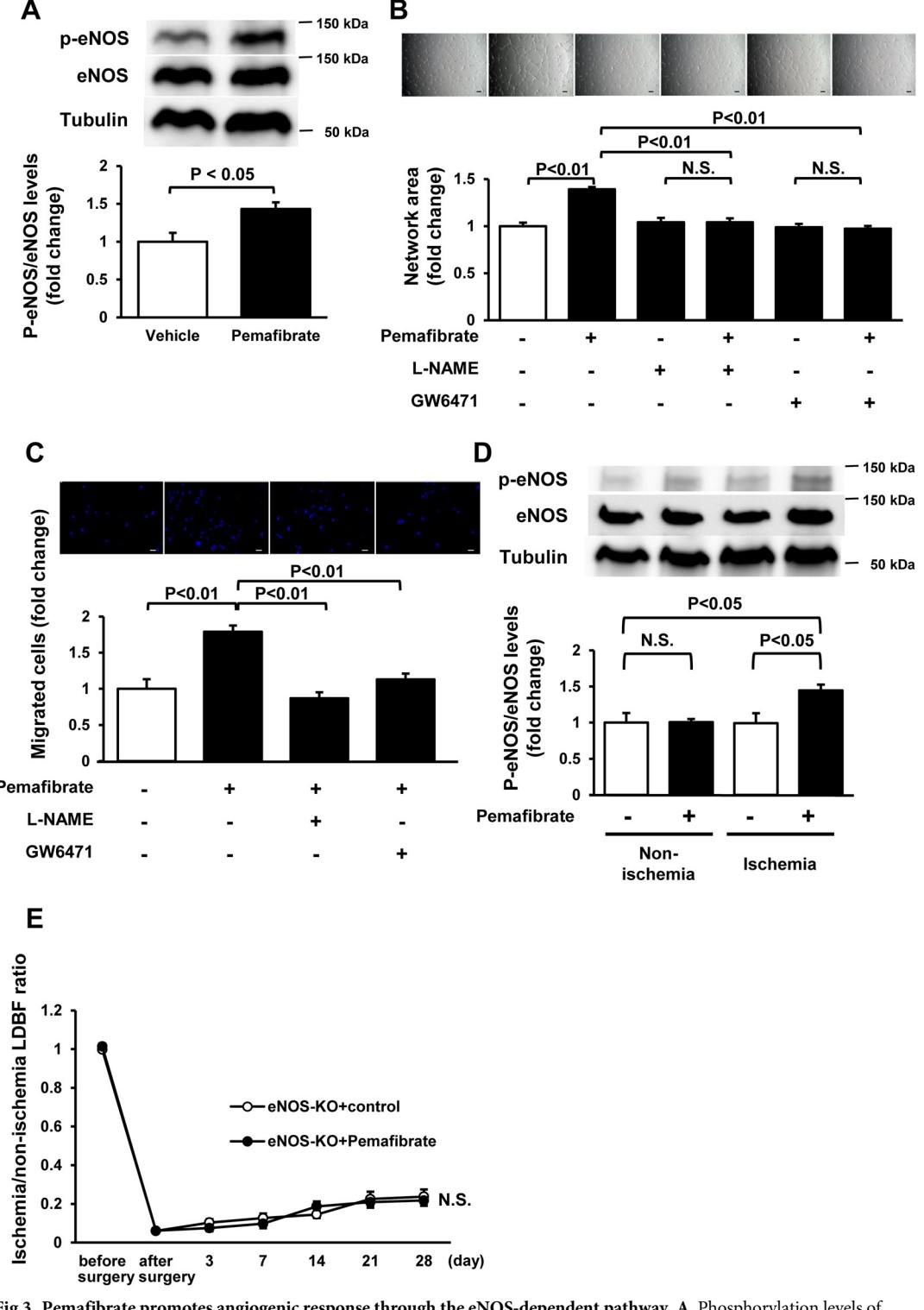

**Fig 3. Pemafibrate promotes angiogenic response through the eNOS-dependent pathway. A**. Phosphorylation levels of eNOS in HUVECs after treatment with pemafibrate. Upper panels show the representative blots of phosphorylated eNOS (P-eNOS), eNOS and α-tubulin (Tubulin) at 1 h after treatment with pemafibrate (10 nM) or vehicle. Lower panel shows the quantitative analysis of phosphorylation levels of eNOS relative to Tubulin. N = 4 in each group. **B and C**. Involvement of eNOS and PPARα in pemafibrate-stimulated enhancement of network formation (B) and migration (C) of HUVECs. HUVECs were pretreated with the NOS inhibitor, L-NAME (500 μM) or PPARα specific inhibitor, GW6471 (50 μM), followed by stimulation with pemafibrate (10 nM) or vehicle. N = 8 in each group (B). N = 4 in each group (C). Scale bars show 200 μm.

**D**. Phosphorylation of eNOS in ischemic limb of WT mice fed pemafibrate or control diet. Upper panels show the representative blots of P-eNOS, eNOS and Tubulin at day 28 after surgery. Lower panel shows the quantitative analysis of phosphorylation levels of eNOS relative to Tubulin. N = 7 in each group. **E**. Effect of pemafibrate administration on blood flow recovery after ischemia in eNOS knockout (eNOS-KO) mice. Quantitative analysis of the ischemic/non-ischemic LDBF ratio of eNOS-KO mice treated with pemafibrate or control diet is shown. N = 10 in each group.

that increased levels of FGF21 by pemafibrate treatment could contribute to an increased blood flow recovery of WT mice.

Finally, we evaluated whether FGF21 modulates eNOS phosphorylation in vitro and in vivo. Treatment with FGF21 protein increased eNOS phosphorylation in HUVECs compared with control (Fig 6A). NOS inhibition by pretreatment with L-NAME blocked the stimulatory effects of FGF21 on endothelial cell differentiation, migration and proliferation (Fig 6B, 6C and 6D). Ad-FGF21 treatment significantly increased phosphorylation levels of eNOS in ischemic muscle, but not in non-ischemic muscle in WT mice compared with Ad-βgal treatment (Fig 6E). Furthermore, Ad-FGF21 did not affect blood flow recovery after surgery in eNOS-KO mice (Fig 6F). These data indicate that FGF21 promotes endothelial cell function and ischemia-induced revascularization through the eNOS-dependent mechanism.

## Discussion

This study provides the first evidence that a novel selective PPARα agonist, pemafibrate promotes endothelial cell function and revascularization under conditions of ischemia. Systemic administration of pemafibrate enhanced blood flow recovery and capillary density in ischemic limbs of WT mice. Treatment with pemafibrate stimulated network formation, migration and proliferative activity of cultured endothelial cells under conditions of normoxia or hypoxia. The stimulatory effects of pemafibrate on endothelial cell function were abolished by PPARα inhibition, indicating that pemafibrate can modulate endothelial behavior in a PPARα dependent manner. Importantly, although FGF21 is a target gene of PPARα, pemafibrate did not affect FGF21 expression in cultured endothelial cells. Thus, it is likely that pemafibrate can affect endothelial cell function in a FGF21 independent manner. These data suggest that pemafibrate can directly modulate endothelial behavior via a PPARα signaling mechanism that is independent of FGF21 induction. Pemafibrate administration also led to significant increases in hepatic expression level of FGF21 and plasma level of FGF21. Systemic administration of FGF21 enhanced ischemia-induced revascularization in WT mice. Treatment of endothelial cells with FGF21 promoted network formation, migratory activity and growth. Thus, it is likely that pemafibrate stimulates revascularization process after ischemia through at least two mechanisms: direct modulation of endothelial cell behavior and enhancement of pro-angiogenic factor FGF21-mediated endothelial cell function.

FGF21 is an endocrine factor that is expressed in several tissues including liver and skeletal muscle [29]. In the present study, treatment with pemafibrate robustly increased FGF21 expression in the liver and plasma levels of FGF21. In contrast, pemafibrate had no effects on FGF21 expression in skeletal muscle tissue, consistent with a previous report [30]. It has been shown that liver is the major source of circulating FGF21 [31]. Thus, these data suggest that pemafibrate administration contributes to elevation of circulating levels of liver-derived FGF21, which affects the function of endothelium in an endocrine manner. However, Thus, future studies using FGF21 deficient mice or FGF21 inhibitors will be required to clarify whether FGF21 is essential for the pro-angiogenic effects of pemafibrate.

It is well known that eNOS plays a pivotal role in regulation of endothelial function and angiogenic response under normal physiological and ischemic conditions [32–34]. Our data

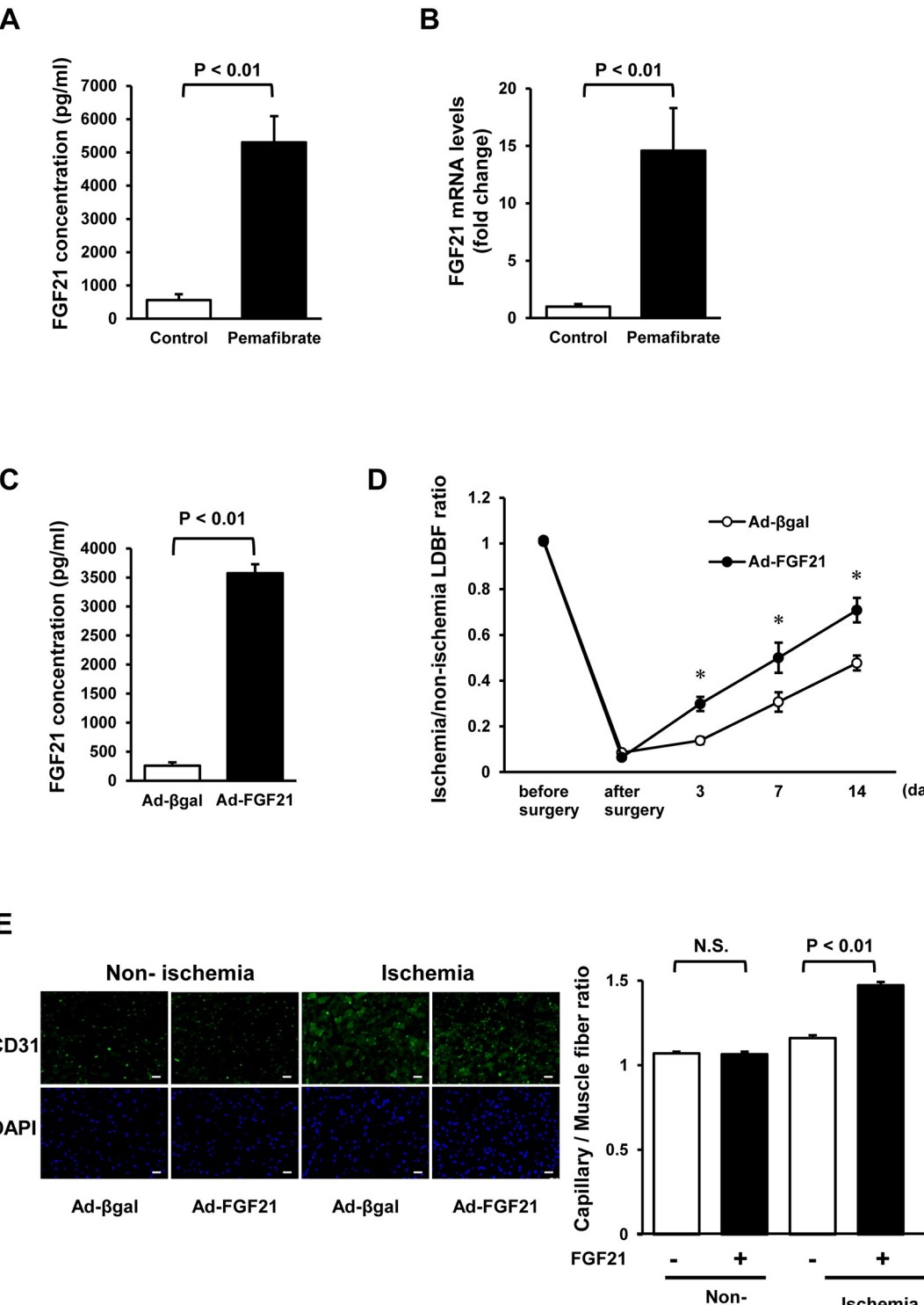

**Fig 4. FGF21 stimulates revascularization in ischemic limbs of WT mice. A**. Plasma concentration of FGF21 of control or pemafibrate-treated WT mice at day 28 after surgery as evaluated by ELISA system. N = 8 in each group. **B**. The mRNA expression of FGF21 in liver of control or pemafibrate-treated WT mice at day 28 after surgery. N = 8 in each group. **C-E**. Effects of FGF21 on blood flow recovery and capillary density in ischemic limb of WT mice. (C) Plasma FGF21 concentration of WT mice at day 3 after treatment with adenoviral vectors expressing FGF21 (Ad-FGF21) or control (Ad-βgal) as evaluated by ELISA system. N = 8 in each group. (D) Quantitative analysis of the ischemic/non-ischemic LDBF ratio of WT mice treated with Ad-FGF21 or Ad-βgal is shown. *P<0.05. N = 8 in each group. (E) Left panels show representative immunostaining of non-ischemic

and ischemic muscle tissues with anti-CD31 antibody (green) and DAPI (blue) on postoperative day 14. Right panel shows quantitative analyses of capillary density in ischemic muscles of Ad-FGF21-treated or Ad-βgal-treated WT mice on postoperative day 14. N = 7 in each group. Scale bars show 50 μm.

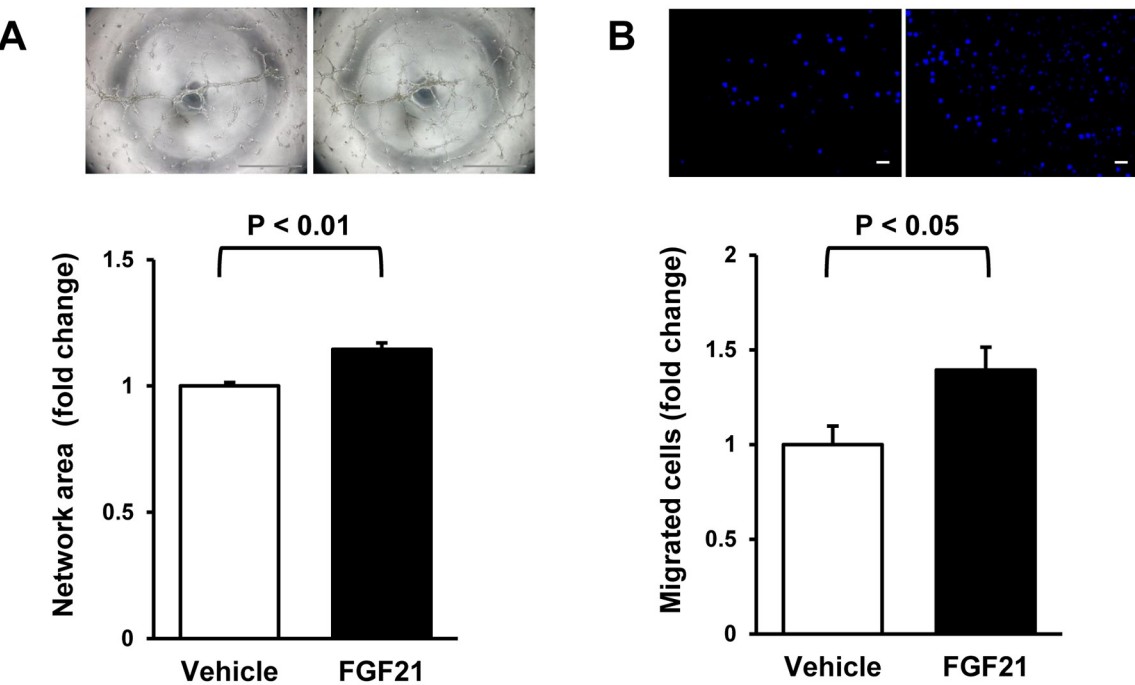

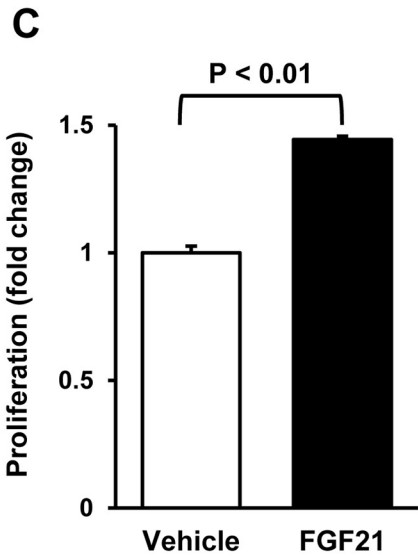

**Fig 5. FGF21 promotes endothelial cell function. A**. Quantitative analysis of network area at 16 h after treatment with FGF21 protein (10 nM) or vehicle. Upper panels show representative photos of network formation of HUVECs. N = 5 in each group. Scale bars show 1 mm. **B**. The number of migrated HUVECs at 8 h after treatment with FGF21 (10 nM) or vehicle. Upper panels show representative photos of DAPI staining of migrated HUVECs. N = 6 in each group. Scale bars show 100 μm. **C**. Proliferative activity of HUVECs at 24 h after treatment with FGF21 (10 nM) or vehicle. N = 10 in each group.

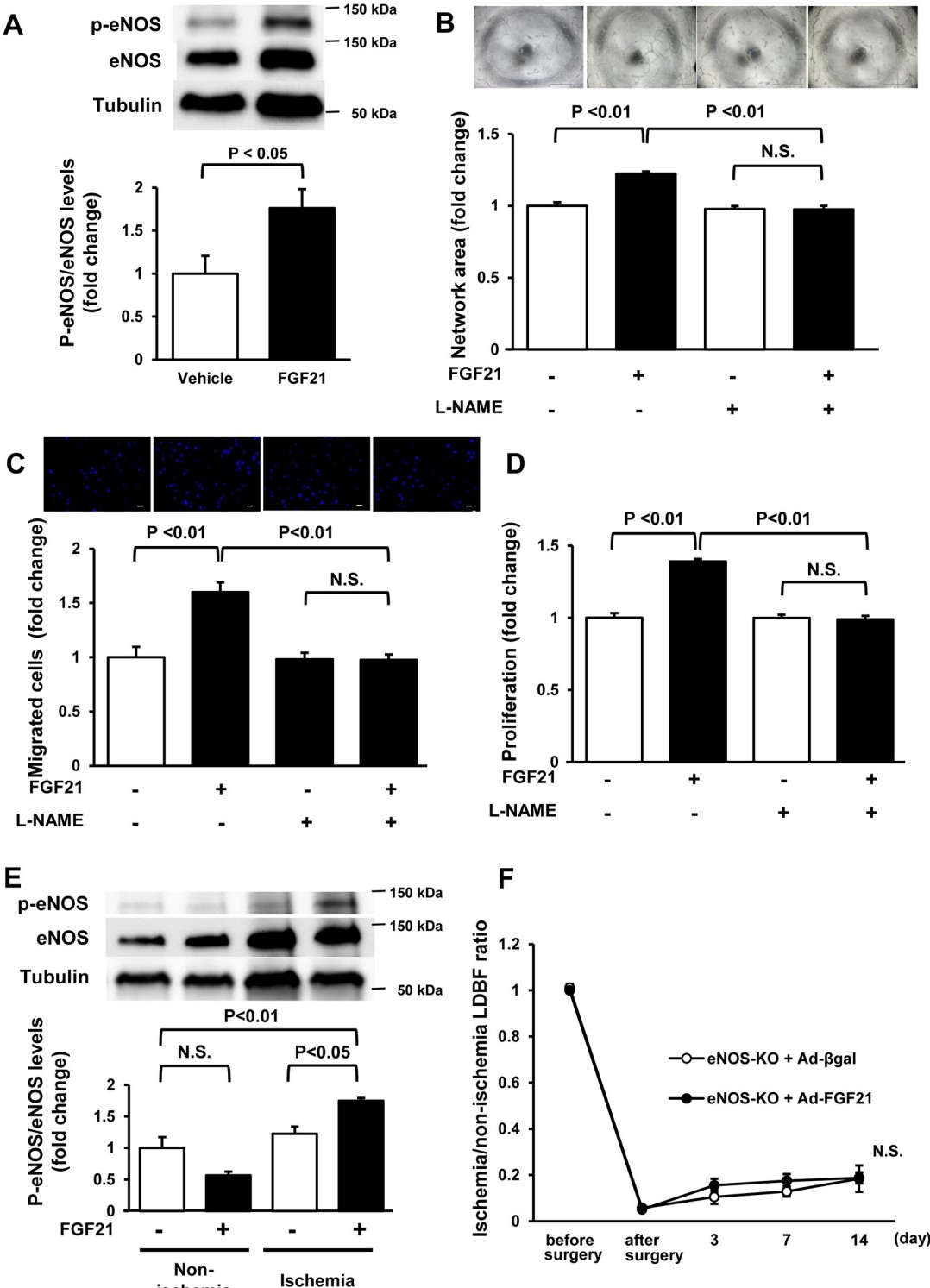

**Fig 6. FGF21 promotes angiogenic response through the eNOS-dependent pathway. A**. Phosphorylation of eNOS in HUVECs after treatment with FGF21. Upper panels show the representative blots of phosphorylated eNOS (P-eNOS), eNOS and α-tubulin (Tubulin) at 1 h after treatment with FGF21 (10 nM) or vehicle. Lower figure shows the quantitative analysis of phosphorylation levels of eNOS relative to Tubulin. N = 5 in each group. **B-D**. Involvement of eNOS in FGF21-induced enhancement of network structure (B), migration (C) and proliferation (D) of HUVECs. HUVECs were pretreated with the NOS inhibitor, L-NAME (500 μM) followed by stimulation with FGF21 (10 nM) or vehicle. N = 8 in each group (B and C), N = 10 in each group (D). Scale bars show 1 mm (B) and 200 μm (C). **E**. Phosphorylation of eNOS in non-ischemic and

ischemic limb of WT mice treated with Ad-FGF21 or Ad-βgal. Upper panels show the representative blots of P-eNOS, eNOS and Tubulin at day 14 after surgery. N = 4 in each group. **F**. Effect of FGF21 on blood flow recovery after ischemia in eNOS knockout (eNOS-KO) mice. Quantitative analysis of the ischemic/non-ischemic LDBF ratio of eNOS-KO mice treated with Ad-FGF21 or Ad-βgal is shown. N = 5 in each group.

showed that pemafibrate promoted endothelial cell network formation and migration in an eNOS-dependent manner. This is consistent with the previous reports showing that fenofibrate activates eNOS in cultured endothelial cells [35, 36]. Our data also showed that FGF21 enhanced endothelial cell function through the eNOS signaling pathway. Furthermore, our data showed that systemic delivery of pemafibrate or FGF21 enhanced eNOS phosphorylation in ischemic skeletal muscle tissues, but not in non-ischemic skeletal muscles. These findings are in agreement with the results that pemafibrate or FGF21 is effective at increasing capillary density only in ischemic tissue but not non-ischemic tissue. However, our in vitro data showed that pemafibrate promoted endothelial cell function under both normoxic and hypoxic conditions. The reason for the discrepancy between in vivo and in vitro effects of pemafibrate on angiogenic response is unknown, and this requires future investigation. Of note, the stimulatory effect of pemafibrate or FGF21 on blood flow recovery in ischemic limbs was abolished under conditions of eNOS deficiency. In addition, pemafibrate dramatically increased circulating levels of FGF21 in mice. Collectively, our data suggest that pemafibrate can promote revascularization in response to ischemia, at least in part, by its ability to promote endothelial cell function through direct and FGF21-mediated activation of eNOS in endothelial cells (S8 Fig).

We previously reported that fenofibrate promotes revascularization in response to ischemia in mice through upregulation of the vasculo-protective adipokine adiponectin [37]. In contrast, the present data demonstrated that pemafibrate had no effects on plasma levels of adiponectin in WT mice. It has also been shown that FGF21 increases adiponectin production in adipose tissue, thereby leading to enhanced levels of circulating adiponectin [38, 39]. Our data showed that pemafibrate did not affect the expression of adiponectin in adipose tissue of mice despite a dramatic increase in circulating levels of FGF21. These findings are consistent with a clinical report showing that pemafibrate increases plasma levels of FGF21 without affecting circulating adiponectin levels [40]. Thus, it is conceivable that the salutary effect of pemafibrate on angiogenic response in vivo is independent of adiponectin.

FGF21 acts as a multifunctional regulator of metabolism and cardiovascular function. FGF21 is reported to improve insulin sensitivity and hypertriglyceridemia [38, 39, 41]. It has also been reported that FGF21 protects against atherosclerosis in a mouse model of atherosclerosis [28, 42]. We have previously reported that FGF21 attenuates adverse cardiac remodeling in mice after myocardial infarction [42]. The present data indicate that FGF21 promotes angiogenic response to ischemic injury. Thus, these data propose that pemafibrate may exert beneficial actions on lipid and glucose metabolism, and cardiovascular disorders partly via upregulation of FGF21 expression.

In conclusion, our present study provides the first evidence that a newly developed SPPARMα, pemafibrate promotes pro-angiogenic response by modulating endothelial cell function. Thus, pemafibrate could be a potentially beneficial drug for prevention or treatment of peripheral arterial disease.

## Supporting information

**S1 Fig. Pemafibrate increases mRNA levels of PPARα target gene, lipoprotein lipase (LPL) in the liver.** N = 3 in each group.
(PDF)

**S2 Fig. Pemafibrate promotes endothelial cell function under hypoxic condition. A**. Endothelial cell network formation after treatment with pemafibrate under hypoxic condition. Upper panels show the representative photos of network formation of HUVECs at 8 h after treatment with pemafibate (10 nM) or vehicle. Lower panel shows the quantitative analysis of network area. N = 8 in each group. Scale bars show 1 mm. **B**. The number of migrated HUVECs at 8 h after treatment with pemafibrate (10 nM) or vehicle under hypoxic condition. Upper panels show the representative photos of DAPI staining of migrated HUVECs. N = 6 in each group. Scale bars show 200 μm. **C**. Proliferative activity of HUVECs at 8 h after treatment with pemafibrate (10 nM) or vehicle. N = 10 in each group.
(PDF)

**S3 Fig. Phosphorylation of eNOS in non-ischemic and ischemic limb of WT mice fed pemafibrate or control diet.** Upper panels show the representative blots of P-eNOS, eNOS and Tubulin at day 7 after surgery. Lower panel shows the quantitative analysis of phosphorylation levels of eNOS relative to eNOS. N = 4 in each group.
(PDF)

**S4 Fig. Plasma concentration of FGF21 of control or pemafibrate-treated WT mice at day 7 after operation as evaluated by ELISA system.** N = 5 in each group.
(PDF)

**S5 Fig. Effects of pemafibrate on FGF21 expression in skeletal muscle and endothelial cells. A and B**. Pemafibrate did not affect the expression of FGF21 in non-ischemic and ischemic skeletal muscle at day 7 (A) and day 28 (B) after surgery. N = 8 in each group (A). N = 5 in each group (B). **C**. Treatment of HUVECs with pemafibrate had no effects on FGF21 expression. N = 5 in each group.
(PDF)

**S6 Fig. Effect of pemafibrate on plasma adiponectin (APN) and adipose tissue APN mRNA levels. A**. Plasma concentration of APN in WT mice fed control or pemafibrate diet. **B**. The mRNA expression of APN in epididymal fat tissue of WT mice fed control or pemafibrate diet. N = 8 in each group.
(PDF)

**S7 Fig. FGF21 levels in plasma and skeletal muscle after treatment with Ad-FGF21 or Ad-βgal. A**. Plasma concentration of FGF21 in WT mice at day 17 after Ad-FGF21 or Ad-βgal administration as evaluated by ELISA system. N = 8 in each group. **B**. The mRNA levels of FGF21 in ischemic skeletal muscle at day 3 (N = 5 in each group) and day 17 (N = 8 in each group) after Ad-FGF21 or Ad-βgal administration.
(PDF)

**S8 Fig. Proposed scheme of the possible mechanisms by which pemafibrate modulates endothelial cell function and revascularization process.** Pemafibrate directly activates eNOS signaling pathway in endothelium. Pemafibrate treatment leads to increases in hepatic FGF21 expression and circulating FGF21 levels, which in turn promote eNOS activation in endothelium. These two pathways are involved in regulation of endothelial cell function and revascularization.
(PDF)

**S1 Raw Images.**
(PDF)

## Acknowledgments

We would like to thank Yoko Inoue and Minako Tatsumi for technical assistance.

## Author Contributions

**Conceptualization:** Koji Ohashi, Toyoaki Murohara.

**Data curation:** Hiroshi Kawanishi, Koji Ohashi, Hayato Ogawa, Naoya Otaka, Noriyuki Ouchi.

**Formal analysis:** Hiroshi Kawanishi, Koji Ohashi, Noriyuki Ouchi.

**Funding acquisition:** Koji Ohashi, Noriyuki Ouchi.

**Investigation:** Hiroshi Kawanishi, Koji Ohashi, Hayato Ogawa, Naoya Otaka, Tomonobu Takikawa, Lixin Fang, Yuta Ozaki, Mikito Takefuji.

**Methodology:** Hiroshi Kawanishi, Koji Ohashi, Naoya Otaka, Mikito Takefuji, Noriyuki Ouchi.

**Project administration:** Koji Ohashi, Toyoaki Murohara.

**Supervision:** Koji Ohashi, Noriyuki Ouchi.

**Validation:** Koji Ohashi, Noriyuki Ouchi.

**Writing – original draft:** Hiroshi Kawanishi, Koji Ohashi, Noriyuki Ouchi.

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
