## [Decision Letter · Decision Letter 0]

18 Mar 2020

PONE-D-20-03337

A novel selective PPARα modulator, pemafibrate promotes ischemia-induced revascularization through the eNOS-dependent mechanisms

PLOS ONE

Dear Dr. Koji Ohashi

Thank you for submitting your manuscript to PLOS ONE. After careful consideration, we feel that it has merit but does not fully meet PLOS ONE’s publication criteria as it currently stands. Therefore, we invite you to submit a revised version of the manuscript that addresses the points raised during the review process.

We would appreciate receiving your revised manuscript by May 02 2020 11:59PM. To enhance the reproducibility of your results, we recommend that if applicable you deposit your laboratory protocols in protocols.io, where a protocol can be assigned its own identifier (DOI) such that it can be cited independently in the future. For instructions see: http://journals.plos.org/plosone/s/submission-guidelines#loc-laboratory-protocols

We look forward to receiving your revised manuscript.

Kind regards,

Masuko Ushio-Fukai, PhD

Academic Editor

PLOS ONE

Journal Requirements:

2. To comply with PLOS ONE submissions requirements, in your Methods section, please provide additional information on the animal research and ensure you have included details on (1) methods of sacrifice, (2) methods of anesthesia and/or analgesia, and (3) efforts to alleviate suffering.

5. Please ensure that you refer to Figure 6 in your text as, if accepted, production will need this reference to link the reader to the figure.

**Comments to the Author**

1. Is the manuscript technically sound, and do the data support the conclusions?

Reviewer #1: Partly

Reviewer #2: Partly

2. Has the statistical analysis been performed appropriately and rigorously? 

Reviewer #1: Yes

Reviewer #2: Yes

3. Have the authors made all data underlying the findings in their manuscript fully available?

Reviewer #1: Yes

Reviewer #2: Yes

4. Is the manuscript presented in an intelligible fashion and written in standard English?

Reviewer #1: Yes

Reviewer #2: Yes

5. Review Comments to the Author

Reviewer #1: The manuscript, a title “a novel selective PPARα modulator, pemafibrate promotes ischemia-induced revascularization through the eNOS-dependent mechanism” by Kawanishi H. et al is very interesting but need to present additional supporting data to prove the hypothesis.

General comments.

The main idea is that pemafibrate increase liver FGF21 and result in increasing circulated-FGF21 in blood which induces phosphorylation of eNOS to enhance ischemia-induced revascularization.

However, authors did not discuss why pemafibrate is effective only in ischemic tissue but not non-ischemic tissue. In addition, it is not clear even though authors suggested at least two mechanisms of pemafibrate at discussion. Please clarify how pemafibrate directly modulates endothelial behavior because authors only presented second effects of pemafibrate via FGF21 in current manuscript.

Authors presented in vitro effects of pemafibrate using HUVECs and showed that pemafibrate increases angiogenic characteristics such as cell migration, network-like tube formation, and proliferation. In vivo experiment, authors showed that pemafibrate was effective only in ischemic-hind limb tissue. It is not consistent between in vitro experiments and in vivo experiments because authors did all in vitro experiments under nomoxic condition.

In addition, authors used 28 days ischemic-limb tissues to show p-eNOS but 1h samples in HUVECs. It is highly possible that FGF21 increase p-eNOS early time point and then the p-eNOS plays a role to enhance revascularization pathway. It might be already finished revascularization at end point 28 days dependent on laser doppler images. Please show all time course for p-eNOS using hind limb ischemic tissues because it is important to check p-eNOS kinetics to enhance revascularization.

Please show ad-FGF21 expression levels in skeletal muscle tissues with time course.

Need to have more detail method section. For example, how authors measured plasma concentration of total choresterol, triglycerol, and glucose.

Please show p-eNOS levels in non-ischemic limb tissues with ischemic tissues in parallel.

Please clarify when authors measured plasma FGF21 concentration after administrating ad-FGF21.

The quantification of network area in Figure 2A is not matched with images. Please change with representative images.

Please show all images for tube formation and migration assays.

Please normalize p-eNOS levels by total eNOS levels instead of tubulin to clarify whether eNOS expression does not change by pemafibrate or FGF21.

Please show FGF21 levels in HUVEC treated with or without pemafibrate.

Please show plasma FGF21 levels with time course in non-ischemic and ischemic tissue.

Please clarify whether FGF21 is an initial key messenger of pemafibrate by showing rescue effect with FGF21 inhibitor or antibody.

Please show effects of pemafibrate in vitro under hypoxic condition to verify in vivo hind limb results.

The conclusion is overestimated because authors did not show any data for lipid metabolism.

Reviewer #2: The manuscript entitled "A novel selective PPARα modulator, pemafibrate promotes ischemia-induced revascularization through the eNOS-dependent mechanisms" is well written. However, the authors need to clarify the following concerns.

1. Throughout the manuscript the authors have referred the figure 6 as figure 5 which is confusing and need serious attention.

2. The IF staining of CD31 in muscle is not much informative. It is very difficult to understand ischemic region without nuclear staining. IHC staining of the same will be more acceptable.

3. The quality of blots are not good enough, such as in figure 3D, 6E.

4. The bar graph for eNOS phosphorylation should be expressed in term of p-eNOS/total eNOS.

5. According to authors, pemafibrate stimulates revascularization through direct effect on endothelial cells. But no mechanism has been provided or predicted. Pemafibrate stimulated revascularization through increased expression of FGF21 does not explain the in-vitro effect.

6. The effect of pemafibrate on FGF21 is well known and the effect of FGF21 on endothelial cell proliferation through eNOS has also been reported. Therefore, the significance of the manuscript can be enhanced if the authors could find the mechanism of direct effect of pemafibrate on ECs.

6. PLOS authors have the option to publish the peer review history of their article (what does this mean?). If published, this will include your full peer review and any attached files.

Reviewer #1: No

Reviewer #2: No

---

## [Author Response · Author response to Decision Letter 0]

1 May 2020

Reviewer #1: 

The manuscript, a title “a novel selective PPARα modulator, pemafibrate promotes ischemia-induced revascularization through the eNOS-dependent mechanism” by Kawanishi H. et al is very interesting but need to present additional supporting data to prove the hypothesis.

Response: We thank the reviewer for these positive comments.

General comments.

The main idea is that pemafibrate increase liver FGF21 and result in increasing circulated-FGF21 in blood which induces phosphorylation of eNOS to enhance ischemia-induced revascularization. However, authors did not discuss why pemafibrate is effective only in ischemic tissue but not non-ischemic tissue. In addition, it is not clear even though authors suggested at least two mechanisms of pemafibrate at discussion. Please clarify how pemafibrate directly modulates endothelial behavior because authors only presented second effects of pemafibrate via FGF21 in current manuscript.

Response: We thank the reviewer for these suggestions. We examined the effects of pemafibrate on HUVEC behaviors in vitro under hypoxic condition. Network formation, migration and proliferation of HUVECs were promoted by pemafibrate treatment even under hypoxic condition. Thus, it is conceivable that pemafibrate treatment can enhance the revascularization and capillary density in ischemic limb of mice. These data are included in Supplemental Figure 2 and Results section (Page 9, line 16-18) in the revised manuscript.

We found that systemic delivery of pemafibrate or FGF21 enhanced eNOS phosphorylation in ischemic skeletal muscle tissues, but not in non-ischemic skeletal muscles. These findings are in agreement with the results that pemafibrate or FGF21 is effective at increasing capillary density only in ischemic tissue but not non-ischemic tissue. These data are included in Figure 3D, Supplemental Figure 3 , Results section (Page 9, line 26 – Page 10, line 3) and Discussion section (Page 13, line 2-6) in the revised manuscript. 

In addition, we found that the stimulatory effects of pemafibrate on network formation and migration of HUVECs were abolished by GW6471, which is a specific inhibitor of PPARα, indicating that pemafibrate can directly modulate endothelial behavior in a PPARα dependent manner. These data are included in Figure 3B, 3C, Result section (Page 9, line 13-18) and Discussion section (Page 11, line 26 – Page 12, line 3) in the revised manuscript.

Authors presented in vitro effects of pemafibrate using HUVECs and showed that pemafibrate increases angiogenic characteristics such as cell migration, network-like tube formation, and proliferation. In vivo experiment, authors showed that pemafibrate was effective only in ischemic-hind limb tissue. It is not consistent between in vitro experiments and in vivo experiments because authors did all in vitro experiments under normoxic condition.

Response: We thank the reviewer for these important suggestions. We examined the effects of pemafibrate on HUVEC behaviors in vitro under hypoxic condition. Network formation, migration and proliferation of HUVECs were promoted by pemafibrate treatment even under hypoxic condition. Thus, it is conceivable that pemafibrate treatment can enhance the revascularization and capillary density in ischemic limb of mice. These data are included in Supplemental Figure 2 and Results section (Page 9, line 16-18) in the revised manuscript.

In addition, authors used 28 days ischemic-limb tissues to show p-eNOS but 1h samples in HUVECs. It is highly possible that FGF21 increase p-eNOS early time point and then the p-eNOS plays a role to enhance revascularization pathway. It might be already finished revascularization at end point 28 days dependent on laser doppler images. Please show all time course for p-eNOS using hind limb ischemic tissues because it is important to check p-eNOS kinetics to enhance revascularization.

Response: As the reviewer pointed out, p-eNOS could increase at earlier time point. Thus, we checked p-eNOS in skeletal muscles at day 7 after surgery. The levels of p-eNOS were higher in ischemic limb of pemafibrate-treated mice than in that of control mice at day 7. This result is now shown in Supplemental Figure 3 and Result section (Page 9, line 26 – Page 10, line 3) in the revised manuscript.

Please show ad-FGF21 expression levels in skeletal muscle tissues with time course.

Response: We thank the reviewer for the useful suggestion. We measured FGF21 expression of skeletal muscles both at day 3 and day 17 after adenoviral vector injection. The mRNA levels of FGF21 increased at skeletal muscle at 3 days after Ad-FGF21 injection compared with control Ad-βgal injection. The mRNA levels of FGF21 also tended to be higher at skeletal muscle at 17 days after Ad-FGF21 injection than Ad-βgal injection, but it was not statistically significant. These results are now included in Supplemental Figure 7B and Result section (Page 10, line 25 – Page 11, line 1).

Need to have more detail method section. For example, how authors measured plasma concentration of total choresterol, triglycerol, and glucose.

Response: We thank the reviewer for these suggestions. We prodived the more detailed description for measurement of total cholesterol, triglyceride and glucose in Materials and Methods sections (Page 5, line 15-16) in the revised manuscript.

Please show p-eNOS levels in non-ischemic limb tissues with ischemic tissues in parallel.

Response: According to the reviewer’s suggestion, we evaluated the p-eNOS levels in non-ischemic limb tissues, too. These results are now shown in Figure 3D and Supplemental Figure 3 in the revised manuscript.

Please clarify when authors measured plasma FGF21 concentration after administrating ad-FGF21.

Response: We measured plasma FGF21 concentration at day 3 after Ad-FGF21 administration. In addition, we evaluated plasma FGF21 concentration at day 17 after Ad-FGF21 administration. These data and points are included in Figure 4C, Supplemental Figure 7A and Result section (Page 10, line 22-25) in the revised manuscript.

The quantification of network area in Figure 2A is not matched with images. Please change with representative images.

Response: According to the reviewer’s suggestion, we added new representative photos of network formation in Figure 2A.

Please show all images for tube formation and migration assays.

Response: According to the reviewer’s suggestions, we added all images for tube formation and migration assay in the revised manuscript.

Please normalize p-eNOS levels by total eNOS levels instead of tubulin to clarify whether eNOS expression does not change by pemafibrate or FGF21.

Response: According to the reviewer’s suggestion, we normalized p-eNOS levels by total eNOS levels in Figure 3A, 3D, 6A, 6E and Supplemental Figure 3 in the revised manuscript.

Please show FGF21 levels in HUVEC treated with or without pemafibrate.

Response: We thank the reviewer for an important suggestion. We checked mRNA levels of FGF21 in the presence or absence of pemafibrate in HUVECs. Treatment with pemafibrate had no effect on the expression levels of FGF21 in HUVECs. These findings are now shown in Supplemental Figure 5C and Result section in our revised manuscript (Page 10, line 15-16).

Please show plasma FGF21 levels with time course in non-ischemic and ischemic tissue.

Response: We thank the reviewer for the suggestion. We measured plasma FGF21 levels at day 7 and day 28 after surgery. Treatment with pemafibrate increased plasma FGF21 levels both at day 7 and day 28 compared with control. These results are shown Supplemental Figure 4, Figure 4A and Result section in a revised manuscript (Page 10, line 9-11). In contrast, there were no difference in the mRNA levels of FGF21 in non-ischemic and ischemic skeletal muscle at day 7 and day 28 after surgery. These data are now shown in Supplemental Figure 5A, 5B and Result section in a revised manuscript (Page 10, line 11-15).

Please clarify whether FGF21 is an initial key messenger of pemafibrate by showing rescue effect with FGF21 inhibitor or antibody.

Response: It is important to clarify whether FGF21 is a key regulator of pemafibrate in angiogenic response in vivo. Thus, we should investigate the contribution of FGF21 to pemafibrate-stimulated revascularization by using FGF21 deficient mice or FGF21 inhibitors. The time limit to revise our manuscript was May 2, which was around 6 weeks later after the first decision. It takes more than 6 weeks to perform these experiments. In addition, we believe that this is beyond the scope of this paper. Thus, future studies using FGF21 deficient mice or FGF21 inhibitors will be required to clarify whether FGF21 is essential for the pro-angiogenic effects of pemafibrate. These points are included in the discussion section in the revised manuscript (Page 12, line 19-21). 

Please show effects of pemafibrate in vitro under hypoxic condition to verify in vivo hind limb results.

Response: We thank the reviewer for these suggestions. We examined the effects of pemafibrate on HUVEC behaviors in vitro under hypoxic condition. Network formation, migration and proliferation of HUVECs were promoted by pemafibrate treatment even under hypoxic condition. These findings are now included in Supplemental Figure 2, Result section (Page 9, line 16-18) and Methods section (Page 7, line 22-25) in the revised manuscript.

The conclusion is overestimated because authors did not show any data for lipid metabolism.

Response: We agree with the reviewer. We deleted the sentence “Clinically, pemafibrate treatment can ameliorate the atherogenic lipid profiles.” We also changed from the phrase “cardiovascular disease” to “peripheral arterial disease” in Discussion section (Page 14, line 9-10). In addition, we deleted the phrase “among patients with dyslipidemia” in conclusion in the abstract. 

 

Reviewer #2: 

The manuscript entitled "A novel selective PPARα modulator, pemafibrate promotes ischemia-induced revascularization through the eNOS-dependent mechanisms" is well written. However, the authors need to clarify the following concerns.

1. Throughout the manuscript the authors have referred the figure 6 as figure 5 which is confusing and need serious attention.

Response: We thank the reviewer for carefully reading our manuscript. We corrected our mistake in the revised manuscript. 

2. The IF staining of CD31 in muscle is not much informative. It is very difficult to understand ischemic region without nuclear staining. IHC staining of the same will be more acceptable.

Response: We thank the reviewer for an important suggestion. We added photos of nuclear staining (DAPI) with CD31 staining in Figure 1C and 4E in the revised manuscript.

3. The quality of blots are not good enough, such as in figure 3D, 6E.

Response: We thank the reviewer for these suggestions. The quality of blots has been improved in Figure 3D and 6E in the revised manuscript.

4. The bar graph for eNOS phosphorylation should be expressed in term of p-eNOS/total eNOS.

Response: According to the reviewer’s suggestion, we evaluated p-eNOS levels in p-eNOS/total eNOS in Figure 3A, 3D, 6A, 6E and Supplemental Figure 3 in the revised manuscript.

5. According to authors, pemafibrate stimulates revascularization through direct effect on endothelial cells. But no mechanism has been provided or predicted. Pemafibrate stimulated revascularization through increased expression of FGF21 does not explain the in-vitro effect.

6. The effect of pemafibrate on FGF21 is well known and the effect of FGF21 on endothelial cell proliferation through eNOS has also been reported. Therefore, the significance of the manuscript can be enhanced if the authors could find the mechanism of direct effect of pemafibrate on ECs.

Response: We thank the reviewer for these important suggestions. we found that the stimulatory effects of pemafibrate on network formation and migration of HUVECs were abolished by GW6471, which is a specific inhibitor of PPARα, indicating that pemafibrate can directly modulate endothelial behavior in a PPARα dependent manner. These data are included in Figure 3B, 3C and Result section (Page 9, line 13-16) in the revised manuscript.

---

## [Decision Letter · Decision Letter 1]

28 May 2020

PONE-D-20-03337R1

A novel selective PPARα modulator, pemafibrate promotes ischemia-induced revascularization through the eNOS-dependent mechanisms

PLOS ONE

Dear Dr. Koji Ohashi

Thank you for submitting your manuscript to PLOS ONE. After careful consideration, we feel that it has merit but does not fully meet PLOS ONE’s publication criteria as it currently stands. Therefore, we invite you to submit a revised version of the manuscript that addresses the points raised by the both reviewers #1 and #2.

Furthermore, please provide the blots or gels include original uncropped blot/gel image data as a supplement file. The blots or gels provided by the authors in the previous review have been already cropped, which are not considered as original ones.

We look forward to receiving your revised manuscript.

Kind regards,

Masuko Ushio-Fukai, PhD

Academic Editor

PLOS ONE

Reviewers' comments:

Reviewer's Responses to Questions

**Comments to the Author**

1. If the authors have adequately addressed your comments raised in a previous round of review and you feel that this manuscript is now acceptable for publication, you may indicate that here to bypass the “Comments to the Author” section, enter your conflict of interest statement in the “Confidential to Editor” section, and submit your "Accept" recommendation.

Reviewer #1: All comments have been addressed

Reviewer #2: (No Response)

2. Is the manuscript technically sound, and do the data support the conclusions?

Reviewer #1: Yes

Reviewer #2: Partly

3. Has the statistical analysis been performed appropriately and rigorously? 

Reviewer #1: Yes

Reviewer #2: Yes

4. Have the authors made all data underlying the findings in their manuscript fully available?

Reviewer #1: Yes

Reviewer #2: Yes

5. Is the manuscript presented in an intelligible fashion and written in standard English?

Reviewer #1: Yes

Reviewer #2: Yes

6. Review Comments to the Author

Reviewer #1: Authors addressed almost reviewer’s comments and all responses are reasonable.

However, reviewer would like to request minor revision as following.

1. It is very important to reproduce authors’ research. Therefore, please provide exact information (company and catalog number, if you modified procedures, please describe it) of experimental assay kits.

2. In the same line, please clarify hypoxic condition (oxygen concentration) and provide a positive control to show that experimental system works well.

3. All images should have scale bars.

4. Almost images are very small. Please provide enlarged images with scale bars.

5. Western blotting images also should have molecular markers.

6. Please carefully check authors’ response. Because authors have wrong response about a part of first comment. Reviewer guess authors just copied and pasted it from response about second comment.

Reviewer #2: Although the authors have tried to answer the queries raised in previous review, still some questions have remain unanswered or not satisfactory.

1. CD31 staining in ischemic and non ischemic muscle is still not convincing. There are no bar graphs in the figures.

2. Since, pemafibrate stimulated EC function in both normoxic and hypoxic condition in vitro, it fails to explain the effect of pemafibrate on ischemic tissue only.

3. Treatment of EC with pemafibrate and GW6471 together could not explain the direct effect of pemafibrate on EC function independent of FGF21 since FGF21 is downstream to PPARα.

7. PLOS authors have the option to publish the peer review history of their article (what does this mean?). If published, this will include your full peer review and any attached files.

Reviewer #1: No

Reviewer #2: No

---

## [Author Response · Author response to Decision Letter 1]

2 Jun 2020

Response to Reviewer 1

Authors addressed almost reviewer’s comments and all responses are reasonable.

However, reviewer would like to request minor revision as following.

Response: We thank the reviewer for the positive comment.

1. It is very important to reproduce authors’ research. Therefore, please provide exact information (company and catalog number, if you modified procedures, please describe it) of experimental assay kits.

Response: We thank the reviewer for the important suggestion. We provided vendor names, catalog numbers of experimental assay kits in our revised manuscript.

2. In the same line, please clarify hypoxic condition (oxygen concentration) and provide a positive control to show that experimental system works well.

Response: We thank the reviewer for the important suggestion. HUVECs were exposed to 5% oxygen concentration by using Anaeropack system. We previously exposed cultured cardiomyocytes (rat neonatal ventricular myocytes) and HUVECs to hypoxia using this system (Ref1-5). Cardiomyocytes led to more than 20% apoptosis after 12 h hypoxia following 24 h normoxia stimulation in this system. We described the information of oxygen concentration and carbon dioxide concentration in the revised manuscript (Page 8, line 4). 

3. All images should have scale bars.

Response: We thank the reviewer for the important suggestion. All photos have scale bars in our revised Figure and Supplemental Figure.

4. Almost images are very small. Please provide enlarged images with scale bars.

Response: We thank the reviewer for the important suggestion. We enlarged photos in the revised Figures.

5. Western blotting images also should have molecular markers.

Response: We thank the reviewer for the important suggestion. We showed molecular weight markers of Western blots in our revised Figures and Supplemental Figures.

6. Please carefully check authors’ response. Because authors have wrong response about a part of first comment. Reviewer guess authors just copied and pasted it from response about second comment.

Response: We regret our wrong response in first comment. The reviewer suggested discussion of the reason why pemafibrate promotes angiogenic response only under ischemic condition. Thus, we discussed this point in Discussion section (Page 13, line 14 -17).

References

1. Ogura Y, Ouchi N, Ohashi K, Shibata R, Kataoka Y, Kambara T, Kito T, Maruyama S, Yuasa D, Matsuo K, Enomoto T, Uemura Y, Miyabe M, Ishii M, Yamamoto T, Shimizu Y, Walsh K, Murohara T. Therapeutic impact of follistatin-like 1 on myocardial ischemic injury in preclinical animal models. Circulation. 2012;126:1728-1738

2. Kataoka Y, Shibata R, Ohashi K, Kambara T, Enomoto T, Uemura Y, Ogura Y, Yuasa D, Matsuo K, Nagata T, Oba T, Yasukawa H, Numaguchi Y, Sone T, Murohara T, Ouchi N. Omentin prevents myocardial ischemic injury through amp-activated protein kinase- and akt-dependent mechanisms. J Am Coll Cardiol. 2014;63:2722-2733

3. Yuasa D, Ohashi K, Shibata R, Mizutani N, Kataoka Y, Kambara T, Uemura Y, Matsuo K, Kanemura N, Hayakawa S, Hiramatsu-Ito M, Ito M, Ogawa H, Murate T, Murohara T, Ouchi N. C1q/tnf-related protein-1 functions to protect against acute ischemic injury in the heart. FASEB J. 2016;30:1065-1075

4. Ohashi K, Enomoto T, Joki Y, Shibata R, Ogura Y, Kataoka Y, Shimizu Y, Kambara T, Uemura Y, Yuasa D, Matsuo K, Hayakawa S, Hiramatsu-Ito M, Murohara T, Ouchi N. Neuron-derived neurotrophic factor functions as a novel modulator that enhances endothelial cell function and revascularization processes. J Biol Chem. 2014;289:14132-14144

5. Joki Y, Ohashi K, Yuasa D, Shibata R, Kataoka Y, Kambara T, Uemura Y, Matsuo K, Hayakawa S, Hiramatsu-Ito M, Kanemura N, Ito M, Ogawa H, Daida H, Murohara T, Ouchi N. Neuron-derived neurotrophic factor ameliorates adverse cardiac remodeling after experimental myocardial infarction. Circ Heart Fail. 2015;8:342-351

Response to Reviewer #2

Although the authors have tried to answer the queries raised in previous review, still some questions have remain unanswered or not satisfactory.

Response: 

1. CD31 staining in ischemic and non ischemic muscle is still not convincing. There are no bar graphs in the figures.

Response: We thank the reviewer for the important comment. We showed the quantitative graphs on the right side of representative CD31 staining.

2. Since, pemafibrate stimulated EC function in both normoxic and hypoxic condition in vitro, it fails to explain the effect of pemafibrate on ischemic tissue only.

Response: The reviewer raised a good point. The reason for the discrepancy between in vivo and in vitro effects of pemafibrate on angiogenic response is unknown. This requires future investigation. These points are included in the discussion section in the revised manuscript (Page 13. Line 14-17).

3. Treatment of EC with pemafibrate and GW6471 together could not explain the direct effect of pemafibrate on EC function independent of FGF21 since FGF21 is downstream to PPARα.

Response: We thank the reviewer for these important suggestions. The stimulatory effects of pemafibrate on endothelial cell function were abolished by PPARα inhibition, indicating that pemafibrate can modulate endothelial behavior in a PPARα dependent manner. Importantly, although FGF21 is a target gene of PPARα, pemafibrate did not affect FGF21 expression in cultured endothelial cells. Thus, it is likely that pemafibrate can affect endothelial cell function in a FGF21 independent manner. These data suggest that pemafibrate can directly modulate endothelial behavior via a PPARα signaling mechanism that is independent of FGF21 induction. These points are included in the discussion section in the revised manuscript (Page 12, line 5 - 12).

---

## [Editor Report · Decision Letter 2]

3 Jun 2020

PONE-D-20-03337R2

A novel selective PPARα modulator, pemafibrate promotes ischemia-induced revascularization through the eNOS-dependent mechanisms

PLOS ONE

Dear Dr. Dr Koji Ohashi

Thank you for submitting your manuscript to PLOS ONE. Although revised manuscript scientifically improved and authors responded to each reviewer's concern satisfactory, supplemental data unfortunately still did not include the "Original and Uncropped and Unadjusted blot/gel image data". Please provide the original gels before cutting and cropping for each figure. For example, there were space or line between blots in Supplement. Please provide the gels/blots without space or lines.

(Instruction from PLOS ONE)

"PLOS ONE now requires that submissions reporting blots or gels include original, uncropped and unadjusted blot/gel image data in addition to complying with our image preparation guidelines described at https://journals.plos.org/plosone/s/figures#loc-blot-and-gel-reporting-requirements. The revised submission should include the raw blot/gel image data for your review, either in Supporting Information or via a public data repository; the Data Availability Statement should indicate where these data can be found. The original blot/gel image data should (1) represent unadjusted, uncropped images, (2) be provided for all blot/gel data reported in the main figures and Supporting Information, and (3) match the images in the manuscript figure(s)"

Please re-submit your revised manuscript within one week. If you will need more time than this to complete your revisions, please reply to this message or contact the journal office at plosone@plos.org. If applicable, we recommend that you deposit your laboratory protocols in protocols.io to enhance the reproducibility of your results. Protocols.io assigns your protocol its own identifier (DOI) so that it can be cited independently in the future. For instructions see: http://journals.plos.org/plosone/s/submission-guidelines#loc-laboratory-protocols

We look forward to receiving your revised manuscript.

Kind regards,

Masuko Ushio-Fukai, PhD

Academic Editor

PLOS ONE

---

## [Author Response · Author response to Decision Letter 2]

9 Jun 2020

Academic editor mentioned like below. According to academic editor's comments, we changed only Raw Images of gel data. 

"Thank you for submitting your manuscript to PLOS ONE. Although revised manuscript scientifically improved and authors responded to each reviewer's concern satisfactory, supplemental data unfortunately still did not include the "Original and Uncropped and Unadjusted blot/gel image data". Please provide the original gels before cutting and cropping for each figure. For example, there were space or line between blots in Supplement. Please provide the gels/blots without space or lines."

---

## [Editor Report · Decision Letter 3]

15 Jun 2020

A novel selective PPARα modulator, pemafibrate promotes ischemia-induced revascularization through the eNOS-dependent mechanisms

PONE-D-20-03337R3

Dear Dr. Koji Ohashi,

We’re pleased to inform you that your manuscript has been judged scientifically suitable for publication and will be formally accepted for publication once it meets all outstanding technical requirements.

Kind regards,

Masuko Ushio-Fukai, PhD

Academic Editor

PLOS ONE

---

## [Editor Report · Acceptance letter]

17 Jun 2020

PONE-D-20-03337R3 

A novel selective PPARα modulator, pemafibrate promotes ischemia-induced revascularization through the eNOS-dependent mechanisms 

Dear Dr. Ohashi:

I'm pleased to inform you that your manuscript has been deemed suitable for publication in PLOS ONE. Congratulations! Your manuscript is now with our production department. 

Kind regards, 

on behalf of

Dr. Masuko Ushio-Fukai 

Academic Editor

PLOS ONE